# Management of Dental Demineralization in a Patient with Complex Medical Conditions: A Case Report and Clinical Outcomes

**DOI:** 10.3390/reports8020039

**Published:** 2025-03-27

**Authors:** Luigi Sardellitti, Enrica Filigheddu, Egle Milia

**Affiliations:** 1Department of Medicine, Surgery and Pharmacy, University of Sassari, 07100 Sassari, Italy; enrica.filigheddu@aouss.it (E.F.); emilia@uniss.it (E.M.); 2Dental Unit, Head and Neck Department, Azienda Ospedaliero Universitaria, 07100 Sassari, Italy

**Keywords:** tooth demineralization, tooth remineralization, oral hygiene, dentin hypersensitivity, preventive dentistry

## Abstract

**Background and Clinical Significance**: Dental demineralization is a multifactorial process influenced by biofilm activity, diet, and systemic conditions. While gastroesophageal reflux disease (GERD) is known for its role in enamel erosion, its contribution to cariogenic processes remains underexplored. Additionally, Brugada syndrome, a genetic arrhythmia disorder, may indirectly affect oral health due to medical complexities and reduced motivation for dental care. This case highlights the management of extensive mineral loss in a patient with GERD and Brugada syndrome, emphasizing the importance of personalized remineralization strategies and interdisciplinary collaboration. **Case Presentation**: A 27-year-old male with Brugada syndrome, treated with a subcutaneous implantable cardioverter defibrillator (S-ICD), presented with widespread enamel demineralization, multiple active carious lesions, and gingival inflammation. Clinical evaluation revealed a high DMFT index (15), significant plaque accumulation, and an oral pH of 5.8, indicating an elevated risk of mineral loss. Poor hygiene habits, frequent sugar intake, and GERD-related acid exposure contributed to his condition. The therapeutic approach included patient education, fluoride-functionalized hydroxyapatite toothpaste and mousse, dietary modifications, and restorative procedures. After 120 days, improvements included enhanced enamel integrity, a reduction in plaque index (from 50% to 25%), and the resolution of gingival inflammation (BOP: 38% to 12%). **Conclusions**: This case underscores the importance of an integrated approach to managing dental demineralization in patients with systemic conditions. The combination of remineralization therapy, behavioral modifications, and structured follow-up yielded significant clinical benefits. Further research is needed to develop standardized protocols for individuals at high risk due to systemic factors affecting oral health.

## 1. Introduction and Clinical Significance

Demineralization is a complex process influenced by biofilm activity, dietary factors, and the oral microenvironment. It represents the primary mechanism behind dental caries, a condition affecting a significant portion of the global population, as highlighted in the Global Oral Health Status Report (WHO, 2022) [1]. Biofilm, a microbial community, metabolizes dietary sugars into acids, leading to mineral loss in dental hard tissues [2,3,4]. This process begins when the pH of the oral environment drops below approximately 5.5, the critical threshold at which hydroxyapatite (HA)—the primary mineral component of enamel and dentin—starts to dissolve [5]. This aligns with Miller’s chemico-parasitic theory, which explains how dietary sugars metabolized by bacteria produce acids that drive demineralization [6].

Saliva plays a central role in mitigating the effects of demineralization through its physical and chemical properties, particularly its buffering capacity [7]. A well-buffered salivary system can neutralize acidic ions and help maintain oral pH levels conducive to remineralization. However, when the acid load exceeds the buffering capacity of saliva, the resulting pH drop promotes demineralization and the formation of cavities [8]. Moreover, various salivary proteins and enzymes protect tooth integrity by preventing the loss of calcium and phosphate ions from the enamel surface or exhibiting antimicrobial activity [9]. Alterations in saliva quantity or quality due to systemic conditions such as radiation therapy, Sjögren’s syndrome, rheumatoid arthritis, or multi-drug therapies are strongly associated with an increased risk of caries development [10]. These changes disrupt the delicate balance between demineralization and remineralization, leaving teeth more vulnerable to damage. This imbalance is further exacerbated in patients with gastroesophageal reflux disease (GERD), where chronic acid exposure accelerates enamel dissolution and demineralization. Unlike bacterial acid production, which is localized to biofilm-covered surfaces, GERD-related acid erosion affects all tooth surfaces, leading to extensive mineral loss and increased susceptibility to hypersensitivity [11,12].

Other systemic conditions, metabolic syndrome (such as diabetes and obesity), deficiencies in dietary micronutrients (e.g., vitamins D and K, calcium, and phosphates), as well as prenatal complications like maternal malnutrition and systemic illnesses during pregnancy, significantly increase susceptibility to the initiation and progression of caries [13,14,15,16,17]. Genetic disorders, such as amelogenesis imperfecta, further compromise tooth mineralization, affecting enamel formation and tooth quality and size, thereby disrupting the overall mineralization process [18].

Similarly, certain genetic syndromes that primarily affect organ systems beyond the oral cavity may also have implications for dental health. One such example is Brugada syndrome, a rare genetic disorder affecting cardiac sodium channels, predisposing individuals to arrhythmias and sudden cardiac death. Brugada syndrome has an estimated prevalence of five per 10,000 individuals, with higher incidence in males and in populations of Southeast Asian descent. The condition is caused by mutations in the SCN5A gene, which encodes a cardiac sodium channel crucial for maintaining normal heart rhythm [19]. Even if no direct association has been established between Brugada syndrome and dental diseases, these patients may have reduced motivation toward oral hygiene and dental care due to the complexity of their medical condition, which includes the need for careful management of stress and anesthetic administration because of their susceptibility to arrhythmias.

Preventive and therapeutic strategies for caries management focus on patient education, behavioral modifications, dietary counselling, and targeted interventions [20]. Fluoride therapy, dental sealants, and non-invasive remineralization treatments, such as casein phosphopeptide-amorphous calcium phosphate (CPP-ACP) and hydroxyapatite nanoparticles, have shown promising outcomes in preventing and arresting demineralization [21,22].

Recent advancements in biomaterials and nanotechnology have led to the development of innovative remineralization agents that not only restore lost minerals but also exhibit antibacterial properties. Hydroxyapatite nanoparticles have emerged as a promising tool in caries management, demonstrating both remineralization potential and antimicrobial activity in clinical studies [23]. Among these, synthetic hydroxyapatite (HA) (Ca_10_(PO_4_)_6_(OH)_2_) has gained significant attention due to its biocompatibility and structural similarity to the natural apatite crystals found in human enamel. This material is incorporated into various oral care products in the form of micro-clusters or nanocrystals to enhance enamel repair and protection [24].

Furthermore, other bioactive materials, such as bioactive glass and peptide-based compounds, are being explored for their ability to support the natural remineralization process and strengthen enamel structure [25].

Despite these advancements, challenges remain in providing effective care for individuals with systemic conditions. Groups such as children, adolescents, the elderly, and those with cognitive impairments or severe systemic diseases are particularly vulnerable to caries due to difficulties in maintaining proper oral hygiene. Overcoming these issues requires an interdisciplinary approach that integrates dental care, nutritional counselling, and systemic disease management to offer tailored and effective solutions.

This case report highlights the management of dental demineralization in a patient with a complex systemic condition, Brugada syndrome, GERD, widespread demineralization, and caries. By combining remineralization therapies with behavioral modifications and patient education, this case report underscores the value of personalized care in improving clinical outcomes.

## 2. Case Presentation

A 27-year-old male patient presented with widespread dental demineralization, reporting increased sensitivity to thermal stimuli and frequent gingival bleeding. He described episodes of spontaneous dental pain, exacerbated by cold beverages and sweet foods. The patient had not undergone regular dental check-ups and had last received professional dental care approximately four years prior for routine prophylaxis.

The patient’s medical history included Brugada syndrome, for which a subcutaneous implantable cardioverter defibrillator (S-ICD) was implanted. Additionally, he had a documented allergy to dust mites, seborrheic dermatitis, bronchial asthma, and gastroesophageal reflux disease (GERD). The patient reported occasional use of Foster (beclomethasone/formoterol) as needed but denied any long-term pharmacological therapy.

Oral hygiene habits were inadequate, characterized by irregular tooth brushing, absence of interdental cleaning, and no use of fluoride-containing products. The patient reported brushing once daily and lacked awareness of the preventive role of fluoride. Dietary habits revealed frequent consumption of fermentable carbohydrates and acidic beverages, particularly energy drinks and soft drinks, with multiple daily intakes. Additionally, he reported episodes of nocturnal bruxism, though no protective measures, such as a night guard, had been adopted.

The oral clinical examination revealed extensive enamel demineralization, primarily affecting the maxillary central incisors. Also, carious lesions were identified on teeth 14, 16, 17, 18, 23, 27, 36, 37, 46, and 47, with several classified as ICDAS 4 or 5, while composite restorations were noted on teeth 11, 12, 13, 21, and 22. The DMFT index was calculated at 15 (D = 10, M = 0, F = 5), indicating a high caries burden.

Gingival assessment demonstrated generalized inflammation with significant plaque and tartar deposits, leading to Bleeding on Probing (BOP) of 38%. The Gingivitis Index (Löe & Silness) was 2.1, indicating moderate to severe gingivitis, with localized areas of increased severity. The Plaque Index (PI) was 50%, and periodontal probing revealed no pockets exceeding 3 mm, excluding significant attachment loss. The Periodontal Screening Index (PSI) was Code 2, confirming the absence of periodontitis but the presence of calculus deposits and gingival inflammation requiring professional prophylaxis. The patient also presented with a deep bite and geographic tongue.

No radiographic signs of periapical pathology were observed. Pulp vitality tests, conducted using cold stimuli, elicited normal responses in all affected teeth, indicating the absence of irreversible pulpal involvement. Salivary analysis revealed a pH of 5.8, consistent with an increased risk of enamel demineralization.

Based on the clinical presentation and diagnostic assessments, the patient was diagnosed with generalized enamel demineralization, multiple active carious lesions, chronic generalized gingivitis, and deep bite with excessive overbite. The demineralization process was attributed to a combination of factors, primarily biofilm activity and dietary influences, but also significantly exacerbated by gastroesophageal reflux disease (GERD). GERD-related acid exposure likely contributed to the widespread mineral loss by lowering the oral pH and increasing enamel dissolution. The differential diagnosis considered included dental fluorosis, which was excluded due to the asymmetric and localized nature of the lesions. Amelogenesis imperfecta was ruled out due to the absence of a family history or generalized enamel hypoplasia. Lastly, early-stage periodontitis was dismissed as there was no evidence of clinical attachment loss or periodontal pocketing beyond 3 mm, confirming that gingival inflammation was restricted to reversible gingivitis.

### 2.1. Phase 1: Patient Education and Introduction of Remineralizing Agents

During the first visit (day 0), after the clinical examination, the patient received comprehensive education on oral hygiene and dietary modifications, emphasizing the importance of mechanical plaque removal and reducing sugar intake. A step-by-step demonstration of the Modified Bass technique was provided, ensuring proper plaque removal while minimizing gingival trauma. The patient was instructed to use a soft-bristled toothbrush and to brush twice daily for at least five minutes. Interdental cleaning was introduced using waxed dental floss.

A low-abrasivity fluoride-substituted hydroxyapatite toothpaste (Curasept Biosmalto Caries Abrasion & Erosion, 1450 ppm F^−^, abrasivity < 56 RDA; Curasept S.p.A, Saronno, Italy) was recommended to enhance remineralization. This toothpaste, formulated with fluorine-substituted hydroxyapatite and bioactive chitosan, was chosen due to its ability to restore mineral density, reduce dentin hypersensitivity, and enhance enamel resistance to acid attacks. The mechanism of action involves the deposition of hydroxyapatite particles onto the enamel surface, mimicking the natural enamel structure and providing a protective barrier against acid erosion [5].

Additionally, the patient was instructed to apply Curasept Biosmalto Caries Abrasion & Erosion remineralizing mousse (1450 ppm F^−^) twice daily. The mousse was applied directly to affected tooth surfaces using a soft toothbrush or fingers and left undisturbed for two minutes before expectorating without rinsing. The formulation of the mousse includes functionalized amorphous calcium phosphate (ACP) combined with fluoride, carbonate, and citrate, designed to enhance the bioavailability of calcium and phosphate ions, promoting enamel remineralization. Unlike the toothpaste, which primarily strengthens the enamel through surface deposition, the mousse delivers a rapid release remineralizing effect, allowing the active ions to penetrate deeper into the enamel lesions, facilitating a faster and more effective remineralization process [26].

The combined use of the hydroxyapatite-based toothpaste and remineralizing mousse aimed to optimize enamel recovery by leveraging both surface reinforcement (toothpaste) and deeper mineral penetration (mousse). Professional supragingival debridement was performed using a sonic scaler and hand instruments to remove plaque and calculus under antibiotic coverage. Dietary counselling reinforced the importance of limiting fermentable carbohydrate intake.

### 2.2. Phase 2: Follow-Up and Clinical Outcomes

The patient underwent six visits, each addressing different aspects of treatment:•Day 14: The restoration of carious lesions in the first quadrant. Cavitated lesions were restored using Enamel plus HRi biofunction composite (Micerium S.p.A, Avegno, Italy). The adhesive technique involved 37% phosphoric acid etching, followed by the application of the Ena Bond Seal adhesive system (Micerium S.p.A, Avegno, Italy) and incremental layering of the composite resin.•Day 35: The restoration of the second quadrant. The patient’s adherence to oral hygiene recommendations was assessed. Professional supragingival and subgingival debridement was performed. The reinforcement of oral hygiene instructions was provided, with additional guidance on optimizing brushing technique.•Day 40: The restoration of the third and fourth quadrants. The patient’s adherence to oral hygiene recommendations was assessed. The reinforcement of oral hygiene instructions was provided.•Day 90: The reinforcement of oral hygiene instructions. A clinical examination was performed to assess the integrity and adaptation of the previously placed restorations, ensuring their proper function and stability.•Day 120: A comprehensive clinical examination was conducted to assess the patient’s progress. A significant remineralization of lesions was observed on the upper anterior teeth, indicating the effectiveness of the remineralization protocol. Gingival health showed marked improvement, with a complete resolution of inflammation. Objective periodontal assessments confirmed a reduction in Bleeding on Probing (BOP) from 38% to 12%, a Gingivitis Index improvement from 2.1 to 1.3, and a Plaque Index (PI) decrease from 50% to 25% (Table 1). The patient was scheduled for regular maintenance visits every six months to ensure long-term stability and prevent the recurrence of oral disease.

A summarized treatment timeline is presented in Table 2.

### 2.3. Photographic and Photometric Analysis

To comprehensively document the patient’s condition and evaluate the effectiveness of remineralization therapy, intraoral photographs at baseline and at 120-day follow-up and photometric analyses were performed. At the initial visit (Day 0), clinical examination revealed extensive demineralization affecting all tooth surfaces, particularly the anterior teeth, along with the presence of a deep overbite and signs of gingivitis, including localized inflammation and bleeding upon probing (Figure 1a). A high-resolution macro photograph was taken providing a detailed close-up of the central incisors at baseline, highlighting specific areas of demineralization and discoloration. Standardization was achieved using a 100 mm macro lens and a mini tripod, maintaining a constant shooting distance of 0.3 m from the patient. The image was captured under controlled lighting conditions. The tooth surface was documented in a dry state to enhance the visibility of demineralization patterns (Figure 1b).

At the 120-day follow-up visit, a full examination showed significant remineralization, particularly on the upper anterior teeth, along with the complete resolution of gingivitis, as there were no signs of inflammation or bleeding upon probing (Figure 2a). A close-up view of the central incisors at this stage emphasized the improved enamel surface and the overall progress in remineralization. This follow-up macro photograph was captured using the same 100 mm macro lens and standardized conditions, ensuring comparability with the baseline images (Figure 2b).

To objectively evaluate changes in enamel mineralization between baseline and follow-up images, a photometric analysis was performed using ImageJ software (version 1.54). Image acquisition was carried out under standardized conditions to ensure data comparability. To eliminate potential interference from color components, images were converted to 8-bit grayscale and subjected to brightness and contrast normalization.

For quantitative analysis, a Region of Interest (ROI) was carefully selected, ensuring that the same area was analyzed in both images for accurate comparison. As minor positional variations between images could affect the results, the TurboReg plugin was employed to perform a rigid body transformation of the follow-up image. This allowed for precise spatial alignment, enabling pixel-by-pixel comparison. Following alignment, key photometric parameters were extracted, including the Mean Gray Value (MGV), which represents the average grayscale intensity within the selected area and serves as an indicator of mineral density, and the Integrated Density (IntDen), which quantifies the total grayscale intensity across the ROI.

Results showed an increase in Mean Gray Value (+3.64%) and Integrated Density (+3.64%) in the post-treatment image, suggesting an enhancement in mineral density.

This standardized approach to photometric and image processing analysis ensures an objective and reproducible assessment of enamel mineralization. By reducing observer bias and allowing for precise pixel-based comparisons, this methodology strengthens the reliability of clinical findings and provides valuable insight into the effectiveness of remineralization treatments (Figure 3).

## 3. Discussion

This case illustrates the complex interplay between systemic conditions, oral environmental factors, and behavioral influences in the development and progression of dental demineralization and caries. While biofilm acid production and dietary habits are primary drivers of demineralization, gastroesophageal reflux disease (GERD) likely played a significant role in exacerbating mineral loss in this patient.

Chronic acid exposure from GERD has been shown to lower salivary buffering capacity and weaken enamel resistance, contributing to an increased risk of demineralization [27,28]. However, the precise mechanisms by which GERD can alter the microbial composition and interact with cariogenic biofilm activity and dietary sugars to accelerate caries progression remain insufficiently understood [29]. In this case, poor oral hygiene, the frequent consumption of fermentable carbohydrates and acidic beverages, and GERD-related acid exposure likely had a cumulative effect, leading to extensive mineral loss and active carious lesions.

The treatment approach prioritized remineralization strategies, behavioral modifications, and minimally invasive restorative interventions. A fluoride-containing hydroxyapatite formulation was selected due to its dual function of promoting enamel resistance and restoring mineral content, complemented by the application of a remineralizing mousse. At the 120-day follow-up, significant improvements were observed, including enhanced enamel remineralization, the resolution of gingivitis, and reduced dentin hypersensitivity. The remineralization protocol adopted in this case included both a fluoride-functionalized hydroxyapatite (HAF) toothpaste and a remineralizing mousse containing amorphous calcium phosphate (ACP) with fluoride and carbonate. The additional use of ACP-based mousse was intended to potentiate the remineralization process by providing bioavailable calcium and phosphate ions in a stabilized form. This dual approach aimed to optimize mineral deposition and prolong the protective effects on demineralized enamel surfaces.

These findings are consistent with previous in vitro and in situ research demonstrating the efficacy of hydroxyapatite-based products in enhancing enamel remineralization and reducing hypersensitivity [5,24,30,31]. While these studies have shown that toothpaste alone can provide significant benefits, the combined application was considered particularly advantageous in this case due to the severity of mineral loss and the need for an intensive remineralization strategy. However, further research is required to compare the long-term outcomes of different remineralization protocols in similar clinical conditions.

Despite the success of the treatment, certain limitations should be considered. As this is a single-case report, it does not aim to provide generalizable conclusions but rather to highlight a clinically relevant scenario that may offer meaningful insights for future research and clinical practice; thus, further studies with larger patient cohorts are needed to validate these findings and establish standardized treatment protocols.

One challenge was the patient’s initial lack of adherence to oral hygiene recommendations, which may have delayed clinical improvements. Although behavioral reinforcement was implemented, future cases could benefit from more structured motivational strategies, such as the digital tracking of oral hygiene habits or periodic reinforcement visits. Additionally, while the remineralization strategy was effective, a longer follow-up period is necessary to assess the durability of the results and determine whether retreatment or additional interventions will be required.

Another limitation is the lack of microbiological analysis to evaluate bacterial shifts in biofilm composition before and after therapy, which could provide further insights into the treatment’s effectiveness at a microbial level. Furthermore, no quantitative dietary assessment was conducted, limiting the ability to objectively evaluate reductions in sugar and acid intake.

The use of hydroxyapatite-based remineralizing agents proved to be a key factor in the patient’s recovery; however, alternative therapies such as casein phosphopeptide-amorphous calcium phosphate (CPP-ACP) or fluoride varnishes could have been considered as additional preventive measures.

Restorative considerations also warrant reflection, particularly regarding the long-term performance of composite restorations in a patient with frequent acid exposure. The restorations were placed using an etch-and-rinse adhesive system to ensure adequate bond strength and marginal adaptation, and follow-up confirmed acceptable retention and function. However, the potential effects of GERD-induced acid exposure on the longevity of these restorations remain unclear, requiring ongoing monitoring for secondary caries and marginal degradation.

This case presents characteristics that set it apart from most studies on remineralization therapies, which primarily focus on carious and white spot lesions or dentin hypersensitivity. While previous research has demonstrated the efficacy of nano-hydroxyapatite-based products in reducing hypersensitivity and promoting remineralization [32] and has shown that fluoride-containing formulations provide superior remineralization following acid attacks [24], the present case involves widespread demineralization associated with systemic conditions, making direct comparisons with the literature challenging. Moreover, although several studies have confirmed the effectiveness of biomimetic hydroxyapatite in remineralizing white spot lesions and reducing hypersensitivity, they did not account for patients with systemic factors contributing to demineralization [33]. Additionally, many studies are conducted in vitro, which limits their direct clinical applicability to complex cases like this one [34,35].

The combination of fluoride-functionalized hydroxyapatite toothpaste and remineralizing mousse aligns with existing evidence while addressing the specific challenges posed by the patient’s condition. This underscores the necessity of a personalized, multidisciplinary approach and highlights the need for further research to refine treatment protocols for medically complex patients.

## 4. Conclusions

This case highlights the importance of a personalized and integrative approach in managing dental demineralization in patients with multiple risk factors, including GERD, poor oral hygiene, and systemic conditions such as Brugada syndrome. The findings support the effectiveness of hydroxyapatite-based remineralization therapies in restoring enamel integrity and reducing dentin hypersensitivity. The case also underscores the need for early diagnosis, patient education, and adherence to structured oral hygiene protocols to prevent disease progression. While this report provides valuable insights, further studies are necessary to validate these findings in larger patient cohorts and establish standardized treatment protocols for high-risk individuals.

## Figures and Tables

**Figure 1 reports-08-00039-f001:**
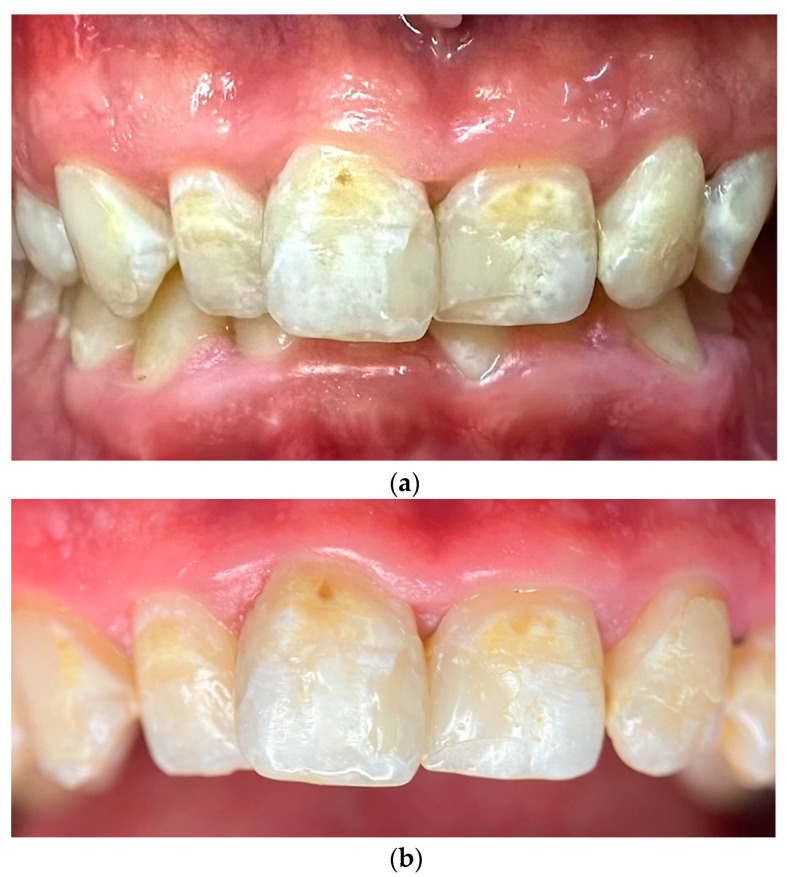
(**a**) Intraoral photograph at baseline (Day 0) showing extensive demineralization and discoloration affecting all tooth surfaces, particularly the upper anterior teeth. The presence of a deep overbite is also evident, as well as the presence of restorations on teeth 11, 12, 21, and 22. (**b**) High-resolution photograph of the central incisors at baseline, highlighting severe demineralization and discoloration. The image was captured under dry conditions to enhance visibility of demineralization patterns.

**Figure 2 reports-08-00039-f002:**
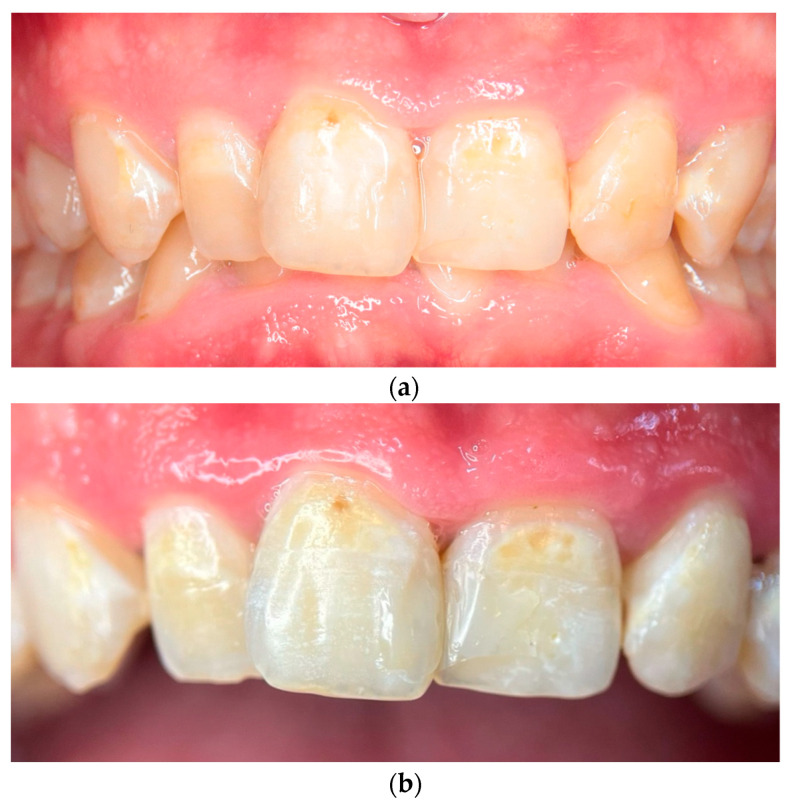
(**a**) Intraoral photograph at the 120-day follow-up showing improvement in enamel remineralization and reduction of discoloration. (**b**) High-resolution close-up of the central incisors at 120-day follow-up, emphasizing the improved enamel surface and successful remineralization. The image was captured under dry conditions to enhance visibility of remineralization patterns.

**Figure 3 reports-08-00039-f003:**
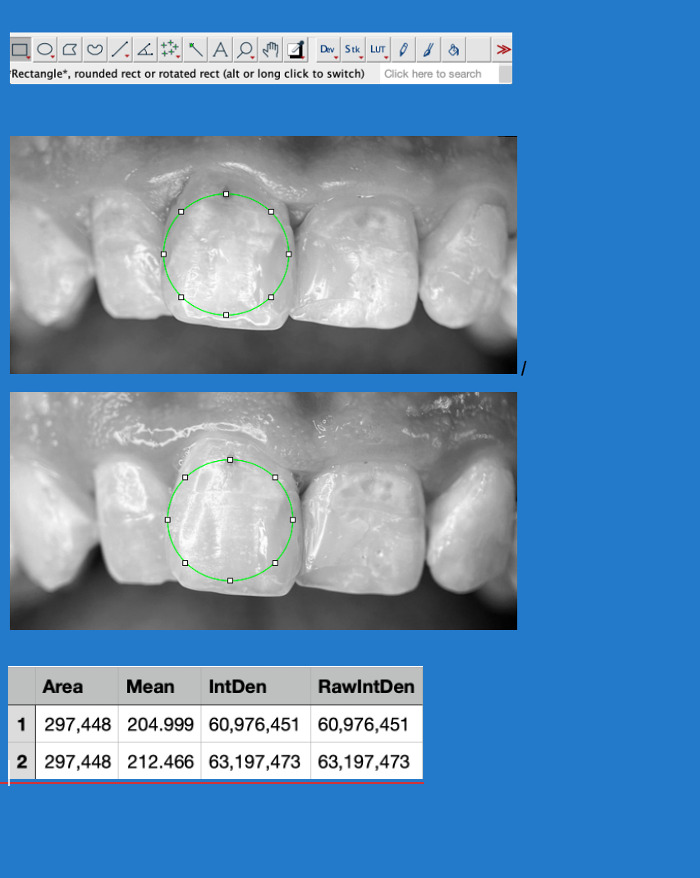
Application of ImageJ software for photometric analysis of enamel mineralization. The selected Region of Interest (ROI), indicated by the green circle, was maintained constant between baseline and post-treatment images to ensure comparability. The white squares represent control points used to define and adjust the ROI within the software. Mean Gray Value (MGV) and Integrated Density (IntDen) were measured to quantify mineralization changes.

**Table 1 reports-08-00039-t001:** Clinical parameters before and after therapy.

Clinical Parameter	Baseline Value	120-Day Follow-Up
Bleeding on Probing (BOP) (%)	38%	12%
Mean Gingivitis Index (Löe and Silness)	2.1 (Moderate to severe gingivitis)	1.3 (Improved gingival health)
Mean Plaque Index (Silness and Löe)	50%	25%
DMFT (Decayed/Missing/Filled Teeth)	15 (D = 10, M = 0, F = 5)	No new lesions

**Table 2 reports-08-00039-t002:** Timeline of clinical interventions.

Event	Details	Date
Visit 1	Education on oral hygiene and dietary counselling. Initiated remineralizing mousse and toothpaste. Supragingival debridement performed.	Day 0
Visit 2	Restoration of carious lesions in the first quadrant using enamel plus HRi biofunction composite. Adhesive technique with 37% phosphoric acid etching and Ena Bond Seal adhesive system.	Day 14
Visit 3	Restoration of second quadrant. Patient’s adherence to oral hygiene recommendations assessed. Professional supragingival and subgingival debridement performed.	Day 35
Visit 4	Restoration of third and fourth quadrants. Patient’s adherence to oral hygiene recommendations assessed. Reinforcement of oral hygiene instructions.	Day 40
Visit 5	Reinforcement of oral hygiene instructions, along with an assessment of the integrity and adaptation of the previously placed restorations.	Day 90
Visit 6	Comprehensive clinical examination.	Day 120

## Data Availability

The data presented in this study are available on request from the corresponding author due to privacy concerns.

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
