# Peer review of "Management of Dental Demineralization in a Patient with Complex Medical Conditions: A Case Report and Clinical Outcomes"

_reports, 2025, doi:10.3390/reports8020039_

Round 1
Reviewer 1 Report
Comments and Suggestions for Authors
- The authors have done dental demineralization study in only patient. How can we conclude about the results from a single patient.
- How Brugada syndrome and asthma contribute to the demineralization? Can the authors provide a control group as the study seems biased.
- What are the components of mousse?
- If only the mentioned toothpaste without the mousse is applied does it work? What are the probabilities?
- What is the mechanism of action for mousse and toothpaste Curasept ?
Author Response
Dear Reviewer
We sincerely appreciate your critical evaluation of our manuscript. Below, we provide responses to each of your concerns and outline the corresponding revisions made.
- The authors have done a dental demineralization study in only one patient. How can we conclude about the results from a single patient?
We acknowledge that a case report inherently presents limitations due to its focus on a single patient. However, the objective of this report was not to draw generalized conclusions but rather to document and analyze an individual case, providing insight into the clinical application of remineralization strategies in a patient with systemic conditions. Our findings contribute to the existing body of knowledge by highlighting a real-world scenario that may guide future studies with larger cohorts. To clarify this, we have explicitly mentioned these limitations in the Discussion section, reinforcing that while the observed outcomes are promising, they should be interpreted as preliminary clinical evidence requiring further validation in larger studies.
- How do Brugada syndrome and asthma contribute to demineralization? Can the authors provide a control group as the study seems biased?
Brugada syndrome does not directly cause dental demineralization. However, it may indirectly impact oral health due to associated medical management, lifestyle factors, and psychological influences. Patients with Brugada syndrome often experience increased stress levels, altered systemic conditions, and the necessity for specific treatments, which could potentially reduce their adherence to oral hygiene protocols. In this case, the patient exhibited poor oral hygiene practices and a high-sugar diet, which were significant contributing factors to his demineralization. Similarly, asthma has been reported to increase caries risk, primarily due to the xerostomic effects of inhaled corticosteroids, although this patient reported only occasional use of such medications.
Regarding the request for a control group, this study was not designed as a comparative trial but rather as a case report, which aims to provide a detailed analysis of an individual patient’s clinical course. The scope of a case report does not include controls, but it serves as a valuable preliminary observation that may encourage future larger-scale comparative studies. We have now clarified this point in the manuscript.
- What are the components of the mousse?
This information has now been explicitly stated in the manuscript to improve clarity.
- If only the mentioned toothpaste without the mousse is applied, does it work? What are the probabilities?
The hydroxyapatite-based toothpaste alone has been shown in previous studies to be effective in enhancing remineralization and reducing dentin hypersensitivity. However, the combination of toothpaste and mousse was chosen in this case to optimize the remineralization process, as the mousse provides a more concentrated release of bioavailable calcium and phosphate, allowing deeper penetration into enamel lesions. While the toothpaste alone may still yield positive effects, the combined approach ensures a more intensive treatment strategy, particularly in cases with extensive mineral loss like the one presented. We have now clarified this rationale in the manuscript.
5. We have now incorporated this information into the manuscript. The toothpaste, which contains fluorine-substituted hydroxyapatite and bioactive chitosan, mimics the natural composition of enamel, reinforcing its structure and increasing its resistance to acid attacks. It facilitates the deposition of hydroxyapatite microclusters, promoting remineralization and reducing dentin hypersensitivity by occluding exposed dentinal tubules.
The mousse, formulated with functionalized amorphous calcium phosphate (ACP), fluoride (1450 ppm), carbonate, and citrate, provides a high concentration of bioavailable calcium and phosphate ions, allowing for deeper penetration into demineralized enamel. This enhances remineralization efficiency and serves as an immediate post-brushing booster, further increasing mineral uptake
Thank you for your time and consideration.
Best regards.
Reviewer 2 Report
Comments and Suggestions for Authors
Dear Authors; I believe that your case report holds significant potential; however, there are several areas that require revision before it can be considered for publication. Highlight any major areas requiring more substantial changes, such as:
1.Case Presentation: A detailed description of the patient’s dental history, clinical examination findings, diagnostic tests.
2.Accurate and complete diagnosis (Definitive diagnosis): Clearly define the condition, outlining its clinical features, diagnostic criteria, and any differential diagnoses considered.
3.Taking high-quality dental photographs is essential for documenting a demineralization, which can be used for diagnosis, treatment planning, and tracking progress over time. The following factors should be considered and standardized:
Camera Settings and Equipment, Focus on Demineralization Areas, Use Standardized Angles, File Management and Labeling, Post-Processing, dryness (dry or wet surface),
Editing; while some minor adjustments (such as brightness or contrast) may change certain features.
4.Tracking progress by using photo management software to track and compare images over time, which can help in assessing the effectiveness of treatment for conditions like demineralization.
- Discussion section: Review of similar cases is meaningful; Comparing the presented case with similar cases in the literature. Explaining commonalities and differences.
Was the treatment approach optimal? Could a different approach have been more effective?
Explaining any challenges faced during treatment is helpful. This might include complications, limitations, or unexpected outcomes.
6.Conclusion section: How does this case contribute to clinical practice? Are there broader implications for treatment, diagnosis, or patient care?
Comments on the Quality of English Languagestyle and quality need improvement
Author Response
Dear Reviewer,
We appreciate your feedback and suggestions, which have helped us to improve our case report. Below, we provide detailed responses to each of your comments and outline the revisions made accordingly.
1. We have expanded the case presentation section by providing a more detailed account of the patient’s dental history, oral hygiene habits, and dietary patterns. Additionally, we have emphasized the systemic factors contributing to the condition, including Brugada syndrome and GERD. A more comprehensive description of the clinical examination findings has been included, particularly highlighting the DMFT index, BOP, Plaque Index, and Gingivitis Index to strengthen the diagnostic assessment.
2. We have clarified the definitive diagnosis, explicitly describing the diagnostic criteria and key clinical features. The differential diagnoses have been discussed, along with the rationale for their exclusion based on clinical and anamnesis data. Additionally, we have highlighted the role of GERD as a contributing factor to the widespread demineralization and provided a more detailed discussion of the ICDAS classification of carious lesions.
3. We have incorporated additional details regarding the photographic methodology used in this study, ensuring standardization in:
camera setup (100 mm macro lens, fixed shooting distance [0.3 m]), lighting and surface conditions (Images were taken under controlled illumination with a dry tooth surface to enhance demineralization visualization), post-processing considerations (minor brightness and contrast adjustments were applied while ensuring the integrity of the diagnostic features).
4. We have included a quantitative photometric analysis using ImageJ software to objectively assess enamel remineralization. Specifically, we implemented:
-Standardized Region of Interest (ROI) to ensure consistent comparison across images.
-TurboReg plugin for precise image alignment to eliminate positional discrepancies.
-Mean Gray Value (MGV) and Integrated Density (IntDen) analysis.
5. We have enriched the Discussion by comparing our case to similar published reports on hydroxyapatite-based remineralization and GERD-related enamel erosion treatments. We specifically addressed similarities and differences between our case and previous literature.
6. We have highlighted the clinical relevance of this case, emphasizing the need for a multidisciplinary approach, the effectiveness of hydroxyapatite remineralization as an alternative to fluoride, and the importance of standardized protocols for GERD-related enamel demineralization in cardiovascular patients.
Improvements in English were made.
Thank you very much for your time and consideration.
Best Regards.
Reviewer 3 Report
Comments and Suggestions for Authors
Dear Authors,
Thank you for the interesting case report of your treatment of a 27-year-old man with Brugada syndrome.
Unfortunately, some points are not yet fully conclusive and comprehensible.
There are some adjustments and additions that should be made.
Introduction
Dear Authors, your focus very much on the oral pH in biofilm and saliva. This is interesting, but there is no concrete link to your case. Please, could you try to link this.
Could you introduce the Brugada syndrome in the introduction (aetiology, epidemiology)? Does the patient have dry mouth or similar, and is there a connection to the introduction?
Case presentation
Could you please add more details to the medical history? Medication (heart medication? asthma spray?), pain, dental experience, etc.?
More photos/radiographs would be very useful for the reader. Could you add more clinical photos and radiographs?
Could you add more clinical findings? Gingivitis index, PSI, DMFT, BOP before and after therapy?
Table 1. Unfortunately, there is no caption. Would you please add one? Furthermore, the therapy performed is only briefly described. It seems as if the patient only had to come twice. Could you please describe how many appointments were attended and what was done?
Could you describe the therapy you performed in much more detail? In the text, you describe it only very briefly. What tooth brushing technique was recommended? Which toothpaste with which fluoride content? What is the advantage of this ‘new toothpaste’? Why was it chosen?
What dental treatments were performed by you? A filling treatment (what material?) can be seen in the second photo. Professional tooth cleaning? Could you please explain this?
How often is the patient called in for follow-up care?
Discussion
Could you critically analyse your therapy? Where were there weaknesses, if any, and what could be improved?
‘While this study shows significant improvements (...)’ You would rephrase this sentence and not speak of a “study”.
Author Response
Dear Reviewer,
We sincerely appreciate your comments and suggestions, which have significantly improved our case report. Below, we provide detailed responses to each of your concerns.
Introduction
You pointed out that our discussion on oral pH in biofilm and saliva lacked a direct link to the case. We have now explicitly connected these factors to the patient’s condition, emphasizing how GERD-induced acid exposure plays a crucial role in enamel demineralization. Furthermore, we have revised the introduction to include a more comprehensive description of Brugada syndrome, outlining its etiology, epidemiology, and potential implications for oral health. While Brugada syndrome itself does not directly cause dry mouth, we have acknowledged that factors such as systemic disease burden, medication use, and stress management could contribute to suboptimal oral hygiene and an increased risk of caries.
Case Presentation
To address your request for a more detailed medical history, we have expanded this section by specifying the patient’s Brugada syndrome treatment, asthma medication usage, pain symptoms, and past dental experiences. We have also added more clinical images to improve visualization of the patient's baseline condition and post-treatment improvements.
Additionally, we have incorporated a more comprehensive report on the clinical indices, specifying the patient’s Gingivitis Index, PSI, DMFT, and BOP before and after therapy. These values have been added to both the text and Table 1.
Treatment Details
You requested further clarification on the treatment protocol. We have now provided additional details on the specific toothbrushing technique recommended (Modified Bass technique), the choice of remineralizing toothpaste (Curasept Biosmalto Caries Abrasion & Erosion, containing 1450 ppm fluoride and hydroxyapatite), and the rationale behind this selection. This particular formulation was chosen due to its dual remineralizing and protective effects on demineralized enamel, as well as its ability to reduce dentin hypersensitivity.
In response to your inquiry about restorative procedures, we have specified that the restorations were performed using Enamel plus HRi biofunction composite (Micerium S.p.A., Avegno, Italy), applied with an etch-and-rinse technique using 37% phosphoric acid and Ena Bond Seal adhesive system. Additionally, we have clarified that professional supragingival debridement was conducted at the first visit using a sonic scaler and hand instruments to improve overall oral hygiene status before commencing restorative interventions. Finally, we have stated that the patient has been scheduled for biannual maintenance visits to ensure long-term stability and prevent recurrence of disease.
Discussion
We have revised the discussion to provide a more critical analysis of the treatment approach, identifying both strengths and areas for improvement. The case demonstrated significant remineralization, reduction in gingival inflammation, and elimination of dentin hypersensitivity, supporting the effectiveness of the selected therapeutic strategy. However, certain limitations were noted, including the lack of microbiological analysis to assess bacterial shifts, the relatively short follow-up period, and the absence of a quantitative dietary assessment.
In response to your concern regarding the wording of the sentence "While this study shows significant improvements (...)", we have modified the text to accurately reflect that this is a clinical case report rather than a study.
Thank you for your time and consideration.
Round 2
Reviewer 1 Report
Comments and Suggestions for Authors
Not satisfied by answer1
Author Response
Dear Reviewer,
Thank you for your feedback and for taking the time to review our manuscript.
We acknowledge that this case report inherently presents limitations due to its focus on a single patient. However, the purpose of this report was not to draw generalized conclusions but rather to provide a detailed clinical documentation and analysis of a specific case involving widespread demineralization associated with systemic conditions.
Case reports are widely recognized as valuable tools for describing rare or complex clinical scenarios that may not be captured by larger studies. They provide preliminary evidence and insights that can guide future research and clinical practice.
We have acknowledged these limitations in the Discussion section of the manuscript, emphasizing that the observed outcomes should be interpreted as preliminary clinical evidence requiring further validation in larger studies.
We hope that this explanation addresses your concern.
Thank you once again for your thoughtful comments.
Reviewer 2 Report
Comments and Suggestions for Authors
Dear authors;
Thank you for submitting the revised version of your manuscript. The changes have significantly improved it. Thank you for your rigorousness and responsiveness to the required revision comments.
Author Response
Dear Reviewer,
Thank you very much for your positive feedback. We appreciate your thorough review and constructive comments, which have greatly helped us improve the quality of our manuscript.
Reviewer 3 Report
Comments and Suggestions for Authors
Dear Authors,
Thank you very much for the comprehensive revision of your manuscript.
It has greatly benefited from this and has improved considerably.
Thank you very much.
Minor improvements should still be implemented:
Page 2 line 80. There is an extra ‘.’ here.
Would you please add the captions to figures 1a,b and 2 a,b.
Author Response
Dear Reviewer,
Thank you very much for your positive feedback. We truly appreciate your acknowledgment of our efforts to improve the manuscript.
We have carefully addressed the minor issues you highlighted:
-
The extra period at Page 2, line 80 has been removed.
-
Captions have been added to Figures 1a, b and 2a, b to provide clear explanations of their content.
Thank you once again for your constructive input.